# Online Equipment Repair Community in Russia: Searching for Environmental Discourse

Olga Zakharova [1,*], Anna Glazkova [1] and Lyudmila Suvorova [2]

1 Green Solutions Lab, University of Tyumen, 625003 Tyumen, Russia; a.v.glazkova@utmn.ru
2 Department of Philosophy, University of Tyumen, 625003 Tyumen, Russia; l.g.suvorova@utmn.ru
* Correspondence: o.v.zakharova@utmn.ru; Tel.: +7-922-269-95-05

**Abstract:** Repair is recognized as an important part of the circular economy and leads to fewer resources being used, less waste, and less emissions generation. The crucial condition for scaling repairs is people's perception of repairs as a significant social practice harmonizing the relationship between society and nature. This paper aims to analyze the key discourses of repair in the content of the posts of the biggest Russian online communities and to identify environmental discourse. These communities specialize in the repair of household appliances, IT, and telecommunications equipment and are organized by independent repairers. We collected all of the posts from the communities that contain textual information in the body of the post. Based on the analysis of the results of the theoretical discourses of repair, we identified four types of repair discourses: environmental, pedagogical, social, and the discourse of the right to repair. We formed lists of keywords corresponding to each discourse type and analyzed posts by computer processing. We concluded that the practice of repair is widespread in specialized online communities, but the content of these communities includes almost no mention of environmental discourse. Social and pedagogical discourses prevail. Based on our research results, we suggested some recommendations for greening and spreading repair.

**Keywords:** repair; repair online community; discourses of repair; social practices; green practices

## 1. Introduction

The recent trend is to minimize the use of resources, as well as waste and emissions generation, due to the growing environmental crisis and climate change, which can be achieved through a circular economy. This implies that waste is revalued as a resource through exchanging, sharing, repairing, reusing, and designing the durability and maintainability of products from the moment of their manufacturing [1]. Thus, producing goods and services in a circular economy is less dependent on extracting resources and emitting waste [2]. To move to a circular economy, governments and corporations of many countries have included this concept in their strategic documents as a new economic model replacing the "take-make-dispose" model [3,4]. However, the researchers note that many specific aspects of interaction between new business models, governmental policies, entrepreneurs, and users' behavior are poorly studied and are not presented in regulatory documents, which reduces the effectiveness of the policy being implemented [5–7]. For example, success in implementing the circular economy depends on businesses adopting circular business models and on users' exhibition of certain behaviors during the acquisition of products during their use and at the end of their useful life [5].

One type of consumer behavior that seeks to extend the use period of items is repair practice, which is recognized as an important part of the circular economy [8]. In this article, we consider repair as "a set of techniques that prolong the life of objects by restoring to them their pragmatic or symbolic function" (p. 61) and as a green practice that enables people to harmonize the relationship between society and nature in the environmental

crisis. The main environmental effects of repair are reducing the use of resources (for example, rare earth minerals) and waste generation by minimizing the ecological footprint of products and supporting product lifetime extension [9].

Despite the positive environmental, economic, and social effects, spreading repair can be difficult in a society of overconsumption because of barriers such as fast fashion [10], strategies of planned obsolescence [11], restrictions on unauthorized repair [12], the perception of repair as an outdated craft [13], or the perception of repair as a sign of poverty [8]. Repairs are spread with the help of enthusiasts, who speak in favor of consumer repair and not only corporate entities. To carry out repairs, independent repairers (for example, the do-it-yourself engineer or the hobby mechanic) and laypersons need some places, tools, skills, knowledge, parts, manuals, and passwords [10,12], access to which in the dominant "throwaway" paradigm is difficult. Apart from all these things for scaling the repair practice, two of the most crucial conditions for scaling repairs and transitioning to a circular economy is people's willingness to get involved in repairs and their perception of repairs as a significant social practice [5].

Repairs can extend to the use or modification of buildings, clothes, and cars, but we will focus on the repair of household appliances, IT, and telecommunications equipment because e-waste is the fastest growing waste stream globally [14], which reached 41.8 million tonnes in 2014 [15]. Most of this waste is not recycled and is not reused, for example, in the European Union less than 40% of E-waste is recycled [16]. A large number of resources are used in electronic production, including rare earth minerals, which then then end up in landfill sites. Moreover, e-waste also contains significant amounts of toxins that are harmful to health, such as mercury, cadmium, chromium, and ozone-depleting chlorofluorocarbons [15]. E-waste results from the use of household appliances, such as washing machines, electric stoves, vacuum cleaners, and toasters, and the use of IT and telecommunications equipment, such as laptops, printers, video cameras, and portable radios, which are used in everyday life and are produced in accordance with strategies of planned obsolescence [12]. Repairs of household appliances and of IT and telecommunications equipment could partially solve the problem of environmental pollution and resource use.

The researchers emphasize the importance of environmental discourse for the organization of repair activities in Europe and the USA [9,11,17,18]. However, no research has been conducted in Russia on the environmental discourse of repair, although the organizers of some repair events declared environmental motives. For example, the first repair cafe in Russia was organized in Perm in 2018, which aimed to provide an opportunity for the participants to save money and reduce waste production [19]. In 2020, the repair cafe was organized by eco-activists in Tula [20]. The eco-activists had environmental goals, such as reducing their carbon footprint by repairing things and reducing consumption without buying new things, as well as social goals, such as gathering creative dwellers in one place and organizing an event that was interesting for both children and retirees. In 2021, an event to promote repair cafes was held in Moscow [21]. In 2022, the Ecocenter organized the first repair cafe in St. Petersburg [22]. The environmental organization ECA issued a guide to popularize the idea of a repair cafe summarizing the experience of such events in different countries [23]. It is worth mentioning that the number of the participants of these repair cafes varied from a few people to three hundred people.

The low prevalence of repair as a green practice can be seen in the results of the investigation of Russian online communities [24]. The practice of repair is mentioned very rarely, according to the investigation conducted on the content of environmental online communities in the Tyumen region (Russia): only three out of six online communities wrote about repair practices, and the number of mentions of repair did not exceed 2% of all mentions of nine green practices considered in the study. Thus, we have previously determined that repair practices are rarely mentioned in environmental online communities in Russia, and repair activities involve only a few people. Recognizing the importance of

repair as a way to solve environmental problems and finding a lack of references to repair as a green practice, we analyze how repair practice exists now and suggest ways to scale it.

This paper aims to analyze the key discourses of repair in the content of the posts of the online community and to identify environmental discourse. We chose the two online repair communities "Soobshhestvo Remonterov" and "Soobshhestvo Remonterov–Pomoshch" organized by independent repairers, such as laypersons, do-it-yourself engineers, and hobby mechanics, and they include about 50,000 people. These communities specialize in repairs of household appliances, IT, and telecommunications equipment. We identified four types of discourses of repair. We formed lists of keywords, which we used to assess the prevalence of the discourses in the posts and discussed the results.

The results of the study will help researchers, volunteers, non-governmental organizations (NGOs), and authorities to understand the motivation of people involved in repairs and develop a strategy for scaling repairs as green practice.

## 2. Theoretical Discourses of Repair

Following Foucault, we understand discourse as an "umbrella" concept, which includes the thematic unity of texts, statements, and a set of ways to formulate any ideas. Discourse analysis provides information about society's attitude to current events, about contradictions and consensuses between social groups, and about the specific peculiarities of social groups. By studying discourse, researchers learn about the social practices in which discourse exists [25]. We share the assumption of a plurality of discourses related to the same social practice and reflecting different aspects. Therefore, the research task is to identify the features of a particular discourse (for example, e.g., keywords) and to find these features in texts.

Through the literature analysis, we identified four key discourses of repair: environmental discourse, pedagogical discourse, social discourse, and the discourse on the right to repair [10,12,26].

### 2.1. Environmental Discourse

Today, environmental discourse underlies theoretical discussions related to repair in Europe and the USA [18,26]. Generally, two environmental issues are affected by repair: increasing the lifespan of materials, which leads to the economy of resources, and counteracting the throw-away culture to reduce waste production [9,17,27]. Most researchers discuss these issues by considering repair as part of the circular economy in the context of the material efficiencies and implementation of the Paris Climate Agreement [7,18]. Therefore, they mention other aspects of the climate agenda. For example, the involvement of large groups in repair will contribute to the maintenance of biodiversity, the prevention of environmental pollution by hazardous waste, and the reduction in the extraction of rare earth metals [8,10]. Moreover, reducing the ecological footprint of local communities and people due to repair and reducing greenhouse gas emissions are widely discussed issues in articles about repairs [11]. Other researchers emphasize the social aspects of the ecological crisis, arguing that repair makes cities attractive and consumption sustainable [10,26].

### 2.2. Pedagogical Discourse

For a long time, people have been repairing things based on skills handed down from generation to generation, but now automation and IT technologies have complicated electronics repair by a layperson [26,28]. Pedagogical discourse is based on the lack of knowledge about how consumer goods are built and function and how to carry out "do it yourself", "do it together" or "do it ourselves" repair [12]. Researchers analyze ways to share skills and knowledge about repairs, such as videos of repairing objects, repair workshops, and repair cafes and parties, as well as repair descriptions of websites [10,28]. On the one hand, researchers emphasize the role of media communications to discuss repairs [28]. Online communities and websites allow people to quickly and widely disseminate information and connect remote like-minded people. On the other hand, researchers

assert the essential value of collective learning, personal presence, and interaction between the human and non-human actors in the repair process [11,26]. The actors who interactare visitors to repair events who recognize a lack of knowledge; repairers who help with repairs; and things that are valuable to their owners [26]. Moreover, the researchers notice that experienced repairers not only transfer knowledge but also learn to teach visitors during interactions [28]. Remarkably, in addition to the knowledge and skills of repair, participation in repair activities increases the overall level of awareness about environmental issues and issues of reducing consumption [29].

*2.3. Social Discourse*

Social discourse goes beyond the consumer identity of the repair participants and includes political, psychological, mental, and gender factors, as well as the factor of the development of local communities [18]. In the political context, repair is discussed as a way of transforming industrial society [30], connecting grassroots initiatives and global politics [28], and settling in the physical space of the local area [8]. Researchers emphasize that participation in repair events contributes to the well-being of low-income groups and equal access to materials, skills, and resources [26]. Publications about repair discuss the topics of designing a future society with limits on raw materials and energy; a post-growth economy; and a post-work society [11,26,31]. The researchers note that repair contributes to the transition from a throw-away culture to an alternative, more sustainable society [11]. The psychological aspects of repair include the experience of self-efficacy; inspiration; and maintaining a sense of safety, stability, and comfort that things give to their owners [9,11,32]. The mental aspects of repair are related to a new way of thinking to get away from overconsumption; they involve creativity and cultivating values such as care, upgrade, repurposing, and appreciation [5,8]. Researchers discuss the gender issues of repair, in which the technological aspects of repair are associated with males and care with females [33]. The researchers note that an important social function of repair is the development of local communities, solidarity through participation in repair cafes, and other events [28]. Therefore, repair activities involve both the local community and enthusiasts such as volunteers, do-it-yourself engineers, and hobby mechanics [26,34].

*2.4. The Discourse of the Right to Repair*

The discourse of the right to repair has a political context and focuses on issues of power (control, influence) and the regulatory framework of repair [9,11,12]. The problem is that owners or users are willing to repair their things not only from corporate entities but also from unauthorized repairs or themselves. However, corporations oppose attempts at unauthorized repair and restrict the right to repair [9]. Researchers discuss the barriers that corporate entities use to monopolize repair, for instance, technological barriers, the voiding of their warranties in case third-party repair, the restriction of access to firmware, error coding, and the use of passwords to block repairs by unauthorized users [12]. Additionally, the repair right requirements focus on free access to repair manuals, spare and interchangeable parts, and the need for maintainability labeling [26,35]. Researchers argue that repair by unauthorized repairs or themselves has a positive social impact, for example, it leads to a reduction in user costs, the development of aftermarket industries, and the development of local repair shops [12]. Moreover, the researchers note that residents of remote areas do not have access to authorized centers and cannot part with their gadgets for a long time; therefore, the right to repair is also necessary for them to ensure social justice [26].

Recognizing the importance of repair as a way to solve environmental problems, we analyze how the repair practice as mentioned in repair online repair communities to suggest ways to scale it as a green practice.

## 3. Methodology

### 3.1. Collecting Data

To analyze the key discourses of repair in the content of social media, we collected posts from the two online repair communities "Soobshhestvo Remonterov" (https://pikabu.ru/community/remont accessed on 10 January 2023) and "Soobshhestvo Remonterov–Pomoshch" (https://pikabu.ru/community/HelpRemont accessed on 10 January 2023). These online communities are hosted on Pikabu, a Russian-speaking information and entertainment network of communities, which is an adapted analog of Reddit. Pikabu works exclusively on the principle of user content. Its owners are not engaged in writing and promoting posts; only registered users can publish a post by creating content or copying from other sources. Unregistered visitors can view the site. Posts and comments can contain formatted texts, as well as attached pictures, gifs, and videos. Two online communities specialize in repairing household appliances and repairing IT and telecommunications equipment; the difference between them lies in the purposes of community members. The purpose of the first community is to share experiences through the publication of reviews of various repairs of appliances, types of equipment, and diagnostics methods. That is why the name of this community reflects its purpose; in Russian, "Soobshhestvo Remonterov" means an association of people who like to repair something. The main content of the community is a description of breakdowns in appliances, equipment, and electronic devices. The purpose of the second community is to publish questions, advice, or requests for help related to repair and to find assistants who can fix problems. This community is called "Soobshhestvo Remonterov–Pomoshch", which means help with repairs. To collect data, we utilized the Python programming language and freely distributed libraries to access the content of the communities. We collected all posts from the communities that contain textual information in the body of the post. The data statistics are summarized in Table 1. The characteristics of the communities are presented as of 10th January 2023.

**Table 1.** Data statistics for "Soobshhestvo Remonterov" and "Soobshhestvo Remonterov–Pomoshch".

| Characteristic | "Soobshhestvo Remonterov" | "Soobshhestvo Remonterov–Pomoshch" |
| --- | --- | --- |
| The publishing period | 6 June 2016–10 January 2023 | 6 February 2017–10 January 2023 |
| The number of subscribers | 39,065 | 12,478 |
| The number of collected posts | 6194 | 8183 |
| The overall number of tokens * | 298.77 | 121.81 |
| The avg length of posts (tokens) | 143.91 | 58.45 |
| The standard deviation of the length of posts | 331.94 | 111.2 |

* The number of tokens is obtained using the Natural Language Toolkit (NLTK) tokenizer [36].

Figure 1 illustrates the distribution of the lengths of posts in the studied communities. As can be seen from the diagrams, the majority of the posts contain no more than 500 tokens. Texts with 100 to 200 tokens prevail.

### 3.2. Forming Lists of Keywords

Based on the analysis of the results of the theoretical discourses of repair, we identified four types of discourses of repair: environmental, pedagogical, social, and the discourse of the right to repair. We formed lists of keywords corresponding to each type, which are presented in Table 2. Since we worked with Russian texts, the table indicates the translations of keywords into English and the original keywords in Russian. In some cases, the English keyword corresponds to several Russian words. This is due to the grammatical

and lexical features of the Russian language, such as the presence of different verbs for expressing grammatical aspects or synonyms.

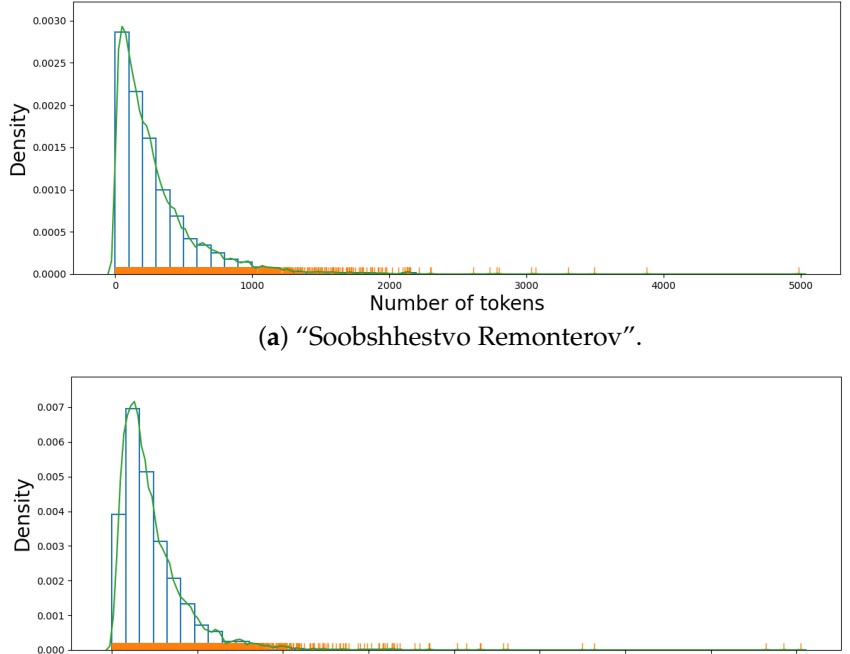

(**a**) "Soobshhestvo Remonterov".

(**b**) "Soobshhestvo Remonterov–Pomoshch".

**Figure 1.** The distribution of the lengths of posts in the communities.

**Table 2.** Keywords for the types of discourses.

| Discourse | Keywords |
|---|---|
| Environmental | circular economy (циркулярная экономика) <br> ecological footprint (экологический/углеродный след) <br> environmentally friendly (экологичный, экологический) <br> greenhouse gas (парниковый газ) <br> hazardous waste (опасные/вредные отходы) <br> material efficiencies (бережливость, беречь) <br> to (not) throw away ((не) выбросить, выбрасывать) |
| Pedagogical | laypersons/hobby mechanics (непрофессионал, любитель) <br> repair description on websites (информация, сайт) <br> skills (навыки) <br> to be able to (уметь) <br> to carry out repair (починить, чинить) <br> to know/to aware (знать) <br> to teach (научить, учить) <br> to transfer knowledge (рассказать, рассказывать) <br> video (видеоролик, видео) <br> workshop (мастер-класс) <br> YouTube |
| Right to repair | access (доступ) <br> error code (код ошибки) <br> password (пароль) <br> regulatory frameworks (закон) <br> repair manuals (инструкция) <br> spare parts/interchangeable parts (запасной) <br> third-party repair (ремонт вне гарантии) <br> to block repairs (блокировка, заблокировать, блокировать) <br> warranties (гарантия) |

**Table 2.** *Cont.*

| Discourse | Keywords |
|---|---|
| Social | appreciation (признательность, нравиться, дорожить)<br>care (забота)<br>come together (собраться, собираться)<br>connecting (взаимодействие, взаимодействовать)<br>creativity (творчество)<br>equality (равенство)<br>help (помощь)<br>hobby (хобби, увлечение)<br>initiatives/actions (инициатива, акция)<br>inspiration (вдохновение)<br>local community (сообщество)<br>males and females (мужчины, женщины)<br>participation (участвовать, участие)<br>politics (политика)<br>repair café (ремонтное кафе)<br>sense of comfort (комфорт)<br>solidarity (солидарность)<br>to discuss (общение, общаться)<br>to help (помочь, помогать)<br>upgrade (апгрейд)<br>volunteer (волонтёр)<br>well-being (благополучие) |

The lists of keywords are used to assess the prevalence of the discourses in the posts. For each keyword, we separately determined the number of occurrences in the texts of each community. Next, we calculated the total number of keywords describing discourses in relative and absolute expression.

*3.3. Data Preprocessing*

To extract keywords, we used the following preprocessing steps for both texts and lists of keywords. We removed special characters, punctuation, and HTML tags and translated texts to lowercase. The texts were lemmatized using the PyMorphy2 library [37]. Finally, we removed commonly used words (stop words). The list of stop words was obtained from NLTK [36] and manually processed. In particular, we excluded the particle "not" ("не") from the list of stop words to find keywords containing the word "not", i.e., "to not throw away".

**4. Results**

Table 3 provides the number of occurrences of the identified keywords in the texts of the two communities. One post may contain several different or coinciding keywords. Some posts may include keywords related to different types of discourse. The Venn diagram (Figure 2) demonstrates the number of texts containing keywords of different discourses. The symbol ∩ indicates the intersection of sets containing keywords of a certain discourse. For instance, $P \cap R$ denotes the intersection of the sets of texts including the keywords corresponding to pedagogical discourse and the discourse of the right to repair.

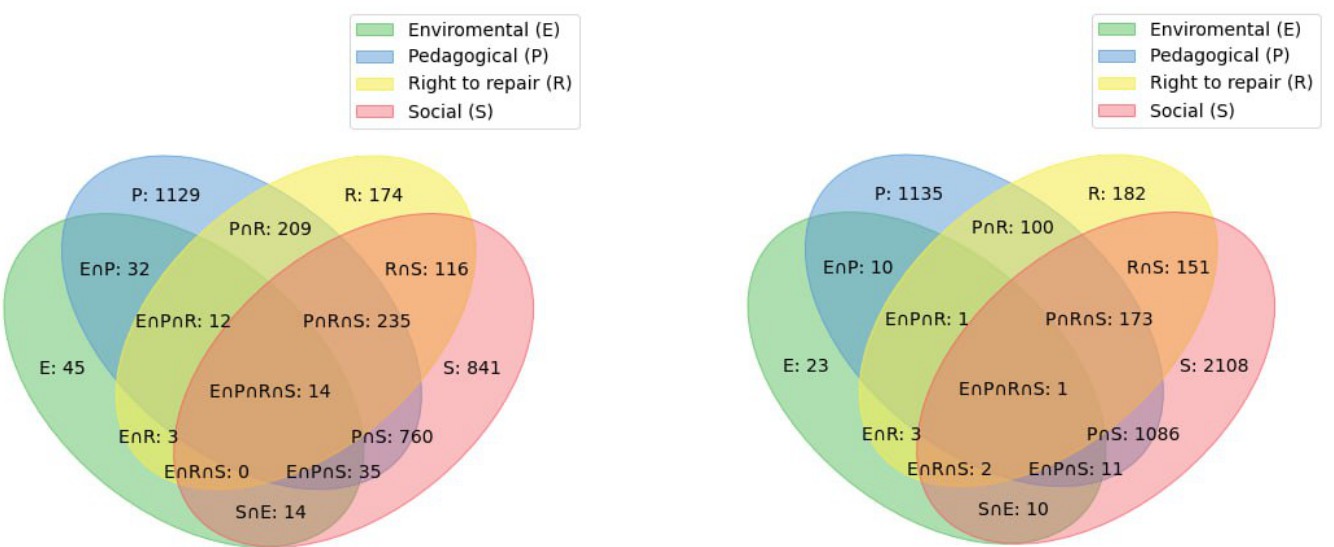

(**a**) "Soobshhestvo Remonterov".

(**b**) "Soobshhestvo Remonterov–Pomoshch".

**Figure 2.** Co-occurrence of four discourses.

**Table 3.** The number of keywords occurrences.

| Discourse | "Soobshhestvo Remonterov" | "Soobshhestvo Remonterov–Pomoshch" |
|---|---|---|
| Environmental | material efficiencies—89<br>to (not) throw away—65<br>environmentally friendly—4<br>circular economy—0<br>ecological footprint—0<br>greenhouse gas—0<br>hazardous waste—0 | to (not) throw away—60<br>material efficiencies—2<br>circular economy—0<br>ecological footprint—0<br>environmentally friendly—0<br>greenhouse gas—0<br>hazardous waste—0 |
| Pedagogical | to know/to aware—990<br>to carry out repair—641<br>video—557<br>repair description<br>on websites— 544<br>to transfer knowledge—488<br>to be able to—169<br>laypersons/<br>hobby mechanics—90<br>YouTube—86<br>skills—36<br>to teach—34<br>workshop—1 | to know/to aware—1104<br>repair description<br>on websites— 572<br>to carry out repair—552<br>video—496<br>to be able to—188<br>skills—74<br>to transfer knowledge—64<br>laypersons/<br>hobby mechanics—47<br>YouTube—41<br>to teach—6<br>workshop—1 |
| Right to repair | warranties—324<br>to block repairs—155<br>repair manuals—145<br>access—143<br>password—82<br>regulatory frameworks—31<br>error code—14<br>third-party repair—9<br>spare parts/<br>interchangeable parts —5 | warranties—223<br>repair manuals—129<br>to block repairs—106<br>password—103<br>access—77<br>error code—17<br>regulatory frameworks—8<br>third-party repair—5<br>spare parts/<br>interchangeable parts —2 |

**Table 3.** *Cont.*

| Discourse | "Soobshhestvo Remonterov" | "Soobshhestvo Remonterov–Pomoshch" |
|---|---|---|
| Social | to help—915<br>help—787<br>hobby—227<br>local community—149<br>come together—122<br>appreciation—110<br>to discuss—75<br>upgrade—52<br>males and females—45<br>participation—33<br>connecting—21<br>politics—15<br>care—11<br>creativity—11<br>initiatives/actions—11<br>sense of comfort—11<br>inspiration—5<br>solidarity—2<br>well-being—2<br>volunteer—1<br>equality—0<br>repair café—0 | to help—2117<br>help—1644<br>local community—313<br>appreciation—42<br>come together—66<br>to discuss—31<br>upgrade—24<br>hobby—22<br>males and females—13<br>participation—13<br>connecting—8<br>initiatives/actions—8<br>politics—6<br>sense of comfort—2<br>creativity—1<br>inspiration—1<br>care—0<br>equality—0<br>repair café—0<br>solidarity—0<br>volunteer—0<br>well-being—0 |

Figure 2 shows the co-occurrence of the four discourses in "Soobshhestvo Remonterov" and "Soobshhestvo Remonterov–Pomoshch". Firstly, in the posts of "Soobshhestvo Remonterov" there is no co-occurrence of environmental discourse, social discourse, and discourse of the right to repair. The environmental and pedagogical discourses in the posts co-occur in less than one percent of the posts of the two discourses. The co-occurrence of the discourse of the right to repair/environmental discourse, and social discourse/environmental discourse, is less than one percent. We see a similar pattern in "Soobshhestvo Remonterov–Pomoshch". Thus, Figure 2 demonstrates the almost complete absence of environmental discourse in community posts. Secondly, in "Soobshhestvo Remonterov" pedagogical and social discourses intersect most often—760 posts, while the intersections of these discourses in "Soobshhestvo Remonterov–Pomoshch" account for 1086 posts. Thirdly, in "Soobshhestvo Remonterov" the intersection of pedagogical and legal discourse occurs twice as often (209 posts) as in "Soobshhestvo Remonterov–Pomoshch" (100 posts). Fourthly, Figure 2 shows that the social discourse in "Soobshhestvo Remonterov–Pomoshch" is more widely represented than in "Soobshhestvo Remonterov".

Figure 3 shows the proportion of texts containing keywords of each discourse for both communities. The proportion of texts $P_D$ of the discourse $D$ is calculated as follows:

$$P_D = \frac{N_D}{T} \times 100\%, \tag{1}$$

$D \in \{environmental, pedagogical, right\ to\ repair, social\}$, $N_D$ is the number of texts containing the keywords of the discourse $D$, and $T$ is the total number of texts in the community.

The absolute number of texts for each discourse is provided in Table 4.

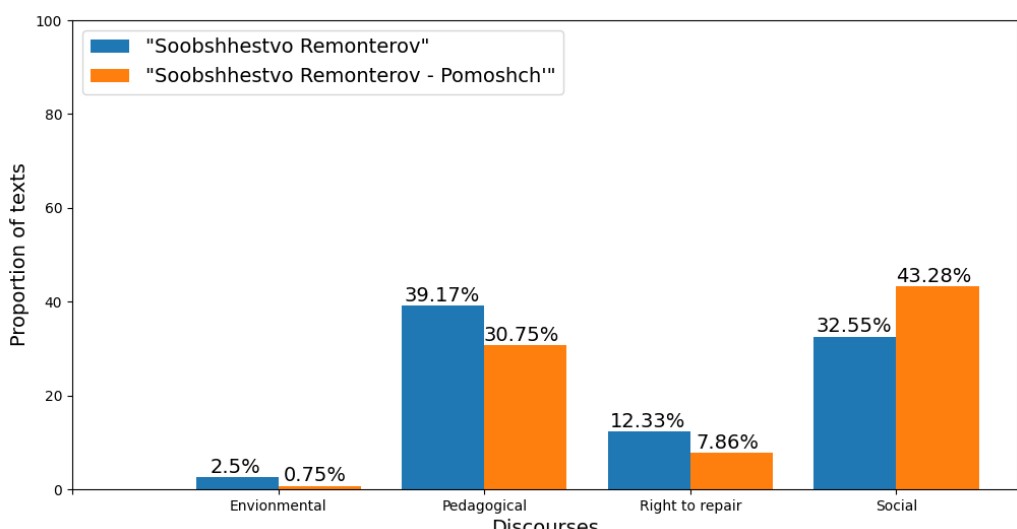

**Figure 3.** The proportion of texts containing the keywords of discourses in each community.

**Table 4.** The number of texts per discourse.

| Community | Environmental | Pedagogical | Right to Repair | Social |
|---|---|---|---|---|
| "Soobshhestvo Remonterov" | 155 | 2426 | 764 | 2016 |
| "Soobshhestvo Remonterov–Pomoshch" | 61 | 2517 | 613 | 3542 |

Figure 3 shows that pedagogical discourse prevails in the content of "Soobshhestvo Remonterov"; it makes up 39.17% of all keywords, and it occurs in 2426 posts (Table 4). In "Soobshhestvo Remonterov–Pomoshch", the keywords of pedagogical discourse make up 30.75% of all keywords and are found in 2517 posts (Table 4). In "Soobshhestvo Remonterov", pedagogical discourse is mainly represented by the words or phrases "to know/to aware"—990; "to carry out repair"—641; "repair description on websites"—544; and "to transfer knowledge"—488 (Table 3). In "Soobshhestvo Remonterov–Pomoshch", these words and phrases also prevail, except for the keyword "to transfer knowledge", which is mentioned only 64 times, compared to "Soobshhestvo Remonterov", where it is mentioned 488 times (Table 3). The keywords "laypersons/hobby mechanics" are more common in "Soobshhestvo Remonterov" (90 times) than in "Soobshhestvo Remonterov–Pomoshch" (47 times). In addition, the keyword "skills" in terms of their lack (absence) are found more often in "Soobshhestvo Remonterov–Pomoshch" (74 times) than in "Soobshhestvo Remonterov" (36 times).

The difference in the meaning of the words and phrases used in the two communities is revealed. In "Soobshhestvo Remonterov", the phrase "to carry out repair" is used as to manage/mismanage, to be capable/not capable, to be necessary, and for the soul. In "Soobshhestvo Remonterov–Pomoshch", the phrase "to carry out repair" is used as to be unable/to fail, and it is also used as part of the questions about who can fix it and how to fix it. The words "to know/to aware" in "Soobshhestvo Remonterov" are used in the following meanings: lack of knowledge, giving knowledge, knowledge will help, knowledge of electronics, and acquiring knowledge. The same words in "Soobshhestvo Remonterov–Pomoshch" are used in the following meanings: lack of skills, lack of knowledge, zero previous level of knowledge, not having knowledge, and lack of knowledge of what to do; basically, they are used in a negative sense. The phrases "repair description/information on websites" are used in "Soobshhestvo Remonterov" in cases when there is no information how to repair something, while in "Soobshhestvo Remonterov–

Pomoshch" these phrases are used in users' requests for repairing. The phrase "to transfer knowledge" in "Soobshhestvo Remonterov" is used in the context of speaking about the repair; the problem; the functioning of appliances, equipment, and electronic devices; and errors in the operation of appliances, equipment, and electronic devices. This phrase in "Soobshhestvo Remonterov–Pomoshch" is used in the context of asking how to properly hand over an item for repair and about errors in the operation of the device. In general, in "Soobshhestvo Remonterov–Pomoshch" the mentioned keywords are often used as part of a question and in "Soobshhestvo Remonterov" as part of an affirmative sentence.

Figure 3 shows that social discourse prevails in the content of "Soobshhestvo Remonterov–Pomoshch"; it makes up 43.28% of all keywords, and it occurs in 3542 posts (Table 4). In "Soobshhestvo Remonterov", the keywords of social discourse make up 32.55% of all keywords and are found in 2016 posts (Table 4). Social discourse is represented by frequently mentioned keywords such as "help/to help", "local community", and "hobby". In "Soobshhestvo Remonterov", the words "help" (787 mentions) and "to help" (915 mentions) are twice as rare as in "Soobshhestvo Remonterov–Pomoshch"—1644 and 2117, respectively (Table 3). The phrase "local community" occurs 313 times in "Soobshhestvo Remonterov–Pomoshch" and is twice as rare (149 times) in "Soobshhestvo Remonterov" (Table 3). The greatest difference between communities is observed in the word "hobby"; in "Soobshhestvo Remonterov", it occurs 227 times, and in "Soobshhestvo Remonterov–Pomoshch" it occurs 22 times (Table 3).

The words "help" and "to help" in "Soobshhestvo Remonterov" are used in the context of providing advice and sharing experience: subscribers help beginners, anyone who needs help. For example, somebody needs a specialist's help, an experienced expert, or any person who is ready to help because he or she has already carried out such repairs. In addition, the words "help" and "to help" are used in the context of describing a tool that can be used to fix problems or to carry out diagnostics of appliances, equipment, and electronic devices. The words "help" and "to help" in "Soobshhestvo Remonterov–Pomoshch" also significantly prevail, but they are used in a different context of asking for help, for example, to repair a device, or asking for help in solving a problem. In addition, subscribers ask for help in determining the causes of problems, diagnostics, asking for advice on where to find a good master, or asking where to find a good service center. Sometimes, subscribers can express doubt whether there is any sense in repairing something because they are unsure if the item in question can be repaired.

The meaning of the phrase "local community" also differs in the two communities. This phrase is used to mean "community of like-minded people" in "Soobshhestvo Remonterov" and "a group of specialists, repair gurus, laypersons, and do-it-yourself engineers who are asked for advice, a hint and help" in "Soobshhestvo Remonterov – Pomoshch". The word "hobby" is often found in "Soobshhestvo Remonterov" because it unites lovers of doing repairs and exchanging experiences; repair is like a hobby of such people. Whereas "Soobshhestvo Remonterov–Pomoshch" aims to find an assistant who can solve the problems with appliances, equipment, electronic devices not carry out repairs by themselves.

The discourse of the right to repair can be much rarer. The keywords of this discourse account for 12.33% (Figure 3) of all keywords, and they are found in 764 posts in "Soobshhestvo Remonterov" (Table 4). In "Soobshhestvo Remonterov–Pomoshch", the keywords account for 7.86% of the total amount of keywords (Figure 3) and are found in 613 posts (Table 4). The words and phrases "to block repairs", "warranties", "access", "repair manuals", and "password" are most often mentioned in Table 3, which corresponds to the content of this discourse described in the section "theoretical discourses of repair".

Environmental discourse in "Soobshhestvo Remonterov" and "Soobshhestvo Remonterov–Pomoshch" is almost excluded. The keywords of this discourse in "Soobshhestvo Remonterov" make up 2.5% of all keywords (Figure 3) in 155 posts (Table 4), and in "Soobshhestvo Remonterov–Pomoshch" they make up 0.75% of all keywords (Figure 3) in 61 posts (Table 4). For example, the phrase "material efficiencies" occurs 89 times.

This phrase is used in the context of economics, not ecology: to save money and to save things. In "Soobshhestvo Remonterov–Pomoshch", the words "to (not) throw away" occur 60 times in the meaning "I don't want to throw away such a thing". In "Soobshhestvo Remonterov", the words "to (not) throw away" are also found, but they have nothing to do with environmental issues. The phrases "circular economy", "greenhouse gas", "hazardous waste", and "ecological footprint" are not found in the discourse.

## 5. Discussion

The study of online communities specializing in repairing household appliances and repairing IT and telecommunications equipment shows that pedagogical and social discourses of repair prevail in the posts' content, which can be explained by not only the historical specificity of repair in Russia but also by international trends in the development of repairs.

Repair has historical origins in traditional Russian peasant society, where things were both produced and maintained locally. Therefore, the majority of the population might have repaired things. Due to late modernization, Soviet people perceived repairing as a common practice during the twentieth century. Moreover, it was harder to get a new item in the Soviet society than to repair an old one in conditions of total shortages. All mass-consumption products were scarce commodities, so they had to serve their owners for a long time [38]. In addition, people generally tend to mend customized things on their own. For example, schoolchildren were taught how to fix or customize furniture and clothes. These skills of repairing toys and bicycles were handed down from generation to generation. Special journals published instructions for repairing household appliances and items. To deal with repair, Soviet families kept everything that might be useful for repairing or that might be repaired.

Sophisticated household appliances could be repaired at specialized household service enterprises. Repair enterprises provided services for sewing, mending, washing, dry-cleaning clothes, repairing shoes, household appliances, televisions, watches, renting various things, and other services. In 1970, there were over 450 types of household services; the most in demand were the individual tailoring and repair of clothes (41%), the repair of shoes (14.3%), and the repair of complex household appliances (14.5%) [39]. In the Soviet Union, the barriers associated with the right to repair were minimized. According to the decree "on measures for the further development of household services for the population", the vocational training of service sector specialists and managers was organized [40]. Any enterprise producing household appliances, electronics, and other consumer goods was obliged to produce spare parts and distribute them throughout the Soviet Union together with repair manuals. Plants produced special equipment for the household service enterprises, and research institutes worked to optimize repair and maintenance processes. A widespread enlightening campaign about the possibilities of repair was carried out among the population through the media, brochures, and short videos. The problem of access to household services for residents of remote settlements was partially solved through the creation of mobile repair centers.

Thus, on the one hand, repair existed as a community and family tradition. On the other hand, repair was developed in the service sector; the state provided the infrastructure, skills, and social meanings of repair. Therefore, historically repair has been associated with pedagogical and social contexts. It is worth mentioning that the traditions of Russian culture can also provide good treatment with respect toward the natural environment through folklore, literature, poetry, philosophy, and emotional support of scientific knowledge [41]. Therefore, the traditions of repair and care for the environment can be harmoniously combined in social practices.

International trends entered Russia along with capitalism at the end of the 20th century. In the post-Soviet period, household services were privatized, and the disposability and availability of electronics, household appliances, clothes, and shoes made repair often unnecessary and sometimes impossible [9,12]. The globalization of fast fashion has become

ubiquitous, and throw-away culture promotes overconsumption [11]. Corporate entities, independent entrepreneurs, and hobby mechanics continue to repair things, but repair skills are not popular in society. The repair practice persists among low-income people as a hobby or as a more economical alternative to buying new goods [5,8,13].

In recent years, the topic of repair has been updated due to the growing environmental crisis [9,17,27]. There are organizations arranging repair activities with environmental slogans, and some people are interested in green initiatives such as repair cafes and online resources for repair [23]. However, repair is not often mentioned even in online communities that organize green practices [24]. In addition, this study showed that environmental discourse is almost non-existent in online repair communities. This can be explained by the fact that most Russians are not concerned with environmental problems as they are satisfied with the environmental situation in their area. According to the surveys of the Russian Public Opinion Research Center (RPORC), every second respondent believes that the environmental situation in their region has not changed for the last two or three years (49%); 17% believe in the improvement of the environmental situation, and 32% of respondents report its deterioration [42]. As for the global environmental situation, according to the RPORC, every second respondent notes the deterioration of the environmental situation in the world over the last two or three years (53%). However, people do not see the connection between the decision to engage in green practices and the ability to influence the global environmental situation. Environmental education and awareness can demonstrate this connection and highlight the contribution of social practices to the ecological situation. According to the RPORC, 38% of respondents believe that to solve environmental problems, it is necessary to increase people's environmental awareness and sense of responsibility. Therefore, pedagogical and social discourse prevailing in online repair communities can be linked to environmental discourse precisely in order to educate the population and to promote more environmentally friendly skills.

The analysis of online communities' content confirms that economic motives for repair are often mentioned. People tend to repair things and ask for advice to make up for a lack of skills when they do not have enough money to buy new things or even for repair. Previous studies examined the relationship between social class and eco-friendly practices, namely, a sense of responsibility for and the ability to protect the earth and concluded that low-income people are less involved in green behavior [43]. To denote the behaviors of people with low income and low cultural capital, E.H. Kennedy and J.E. Givens used the term eco-powerlessness, emphasizing the sense of powerlessness to effect positive environmental change and the tendency to ignore environmental issues. The results of the study by E.H. Kennedy and J.E. Givens explain why people engaged in repairs due to a lack of money rarely mention environmental issues. In addition, their study showed that people with high cultural capital and income demonstrate the highest level of environmental concern and willingness to engage in environmental issues. Therefore, the popularization of repair as a green practice among middle-class people can attract people to repair, even those who still prefer buying a new thing and produce the greatest ecological footprint [44]. Environmental education, as well as initiatives to organize student repair cafes, could be a way to popularize repair as a green practice.

## 6. Conclusions

The study showed a paradoxical situation: the practice of repair is widespread in specialized online communities, but the content of these communities includes almost no mention of environmental discourse. Social and pedagogical discourses prevail. This may be explained, on the one hand, by historical traditions and an established attitude to repairs and, on the other hand, by a low level of environmental concern in society. Thus, repairs in the studied communities are not considered as green practice.

Nevertheless, the disclosure of the ecological potential of this practice is necessary for its scaling. Based on our research, we suggest two ways to do this. The first way is to involve eco-activists in repair. By joining in on repair, eco-activists can bring new

environmental content to this practice. For example, eco-activists can organize events related to the environmental education of communities' subscribers. The second way is for the leaders of the repair communities to organize eco-activities to popularize the environmental agenda among subscribers. For example, it is possible to involve subscribers of the online community in the activities of repair cafes, which are gaining popularity around the world and are closely related to the environmental agenda.

The study showed that the discourses often considered the co-occurrence in the same posts, so the mention of environmental discourse, the discussion of environmental problems, and the discussion of greening society can be combined with more popular social and pedagogical discourses. This finding can be used by leaders of online repair communities and eco-activists who will prepare posts for online repair communities. In addition, we found a lack of awareness of the connection between individual efforts and the global environmental situation. Therefore, education can illuminate which efforts are preferable to overcome the environmental crisis and reduce consumption.

In addition, the study showed which ways could be used to promote the practice of repair minimization, reduce the ecological footprint of products, and support product lifetime extension. Firstly, we believe that the experience of the Soviet Union, which led to the massive spread of repair, can be further studied and used. Regardless of the reasons, the management of repair and its support and promotion were effectively organized. Secondly, the study showed that historical traditions can also serve as a basis for the spread of repair practices, and repair can be promoted as a revival of crafts and family events for the passing down of traditions. These traditions can be combined with other traditions, for example, gratitude for and admiration of nature. The careful preservation of social and historical experience in conditions of dwindling resources and a post–grow economy is an important method for creating new social practices [45,46].

The research results can be used to implement strategies for scaling up the repair practice as a green practice. On the one hand, the results obtained can be the basis for the decisions authorities make regarding the greening of social development. In addition, the results obtained can improve existing programs to support grassroots initiatives by focusing on events such as repair cafes. On the other hand, the actions of eco-activists can be adjusted in accordance with our recommendations. For example, the results of the study will help to engage the leaders of online repair communities in promoting the environmental agenda.

## 7. Limitations and Future Research

The main limitation of the study is the need to translate keywords found in English language texts into Russian. We had difficulty in selecting identical words. In addition, many Russian words have several meanings, which complicates the automatic search for suitable keywords. Additionally, we studied the posts of only two online repair communities, which does not allow us to draw conclusions about the prevalence of repair practices among all online communities and about the prevailing types of repairs. It is necessary to study communities in other social networks that organize other types of repairs in order to assess the prevalence of repair practices and their connection with environmental discourse.

**Author Contributions:** Conceptualization, O.Z.; writing—original draft preparation, L.S.; writing—review and editing, O.Z.; methodology, A.G.; formal analysis, L.S. and A.G.; visualization, A.G.; and project administration, O.Z. All authors have read and agreed to the published version of the manuscript.

**Funding:** This research received no external funding.

**Institutional Review Board Statement:** The ethical review and approval of this study were canceled because no personal data were used.

**Informed Consent Statement:** Informed consent was obtained from all subjects involved in the study.

**Data Availability Statement:** The data presented in this study are available on request from the corresponding author.

**Acknowledgments:** We are grateful to Valeria Evdash (Center for Academic Writing "Impulse", University of Tyumen) for her assistance with the English language.

**Conflicts of Interest:** The authors declare no conflict of interest.

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
