# Peer review of "Online Equipment Repair Community in Russia: Searching for Environmental Discourse"

_sustainability, doi:10.3390/su151712990_

Round 1
Reviewer 1 Report
Thank you for the opportunity to read this paper.
The title of the paper is appropriate for the conttent.
The aim is unclear for me: Do the authors wanted to prove tha the repair practice is not associated with environmental discourse, but it was rather determined by necesity? In my opninion the environmental practice is more important that the environmental disourse. Analyzing text the authors talks only about discourse. How do they measure practice in this article? Is there any indicator? Nevertheless, the idea of repair caffe is constructive.
The authors have to take into account that the linear economic model of development (take-make-dispose pattern) is reaching its limits. The materials are harvested and extracted, then are used to manufacture a product that is sold to consumer, when then discard it when no longer serve its purpose. The companies’ exposure risk is increasing in the same time with the increase of price for materials, associated with supply disruptions. A circular economy would not just ‘buy time’ but also reduce the amount of material consumed to a lower set point.
This is the point in which every company looks for circular economy model of development (prevent-reuse-recycle-recover-dispose pattern) tying to take the advantages of global reverse networks, reorganizing the materials flows, bringing innovations in business models with the help of technology and human kind cooperation.
The authors should analyse all these 5 elements.
The metodology should be detailed and improved. An analysis based on percentages is supperficial.
Author Response
|
Reviewer 1. |
|
|
The aim is unclear for me |
Recognizing the importance of repair as a way to solve environmental problems and finding a lack of references to repair as a green practice, we analyze how repair practice exists now and suggest ways to scale it. We have added this sentence to the introduction (line 87-89 of the new version of the article). The purpose of the article is indicated in lines 90-91. |
|
Analyzing text the authors talks only about discourse. How do they measure practice in this article? Is there any indicator? |
Following Foucault, we understand discourse as an "umbrella" concept that includes the thematic unity of texts, statements, a set of ways to formulate any ideas. Discourse analysis provides information about society's attitude to current events, about contradictions and consensuses between social groups, about the peculiarities of social groups. By studying discourse, researchers learn about social practices in which discourse exists (Dryzek J.S. (2013) The Politics of the Earth: Environmental Discourses. 3rd ed. Oxford: Oxford University Press). We share the assumption of a plurality of discourses related to the same social practice and reflecting different aspects. Therefore, the research task is to identify the features of a particular discourse (e. g., keywords) and to find these features in the texts. For this purpose, we have identified four types of repair discourses. We made lists of keywords that we used to assess the prevalence of the discourses in the posts. We have added this part to the Theoretical discourses of repair (lines 102-110) of the new version of the article). This paper aims to analyze the key discourses of repair in the content of the posts of the online community and to identify the environmental discourse. This is our way of understanding how people evaluate their repair activities. |
|
This is the point in which every company looks for circular economy model of development (prevent-reuse-recycle-recover-dispose pattern) tying to take the advantages of global reverse networks, reorganizing the materials flows, bringing innovations in business models with the help of technology and human kind cooperation. The authors should analyse all these 5 elements. |
We agree that the Circular Economy emerges from resource scarcity and plays an important role in addressing environmental problems, such as reduced resources for production and increased pollution. We described this in the introduction (lines 17-31). In this article we focus on repair. Circular business models increasingly include repair, but our research interest is not related to corporate entities, but to enthusiasts such as volunteers, do-it-yourselfers and hobby mechanics. (lines 91-95). Therefore, the article does not discuss business models, but rather public repair initiatives, such as online repair communities or repair cafes. It is important for us to understand how much community members understand that repairing helps solve environmental problems, so we study what participants write in their posts in the online community. |
|
The metodology should be detailed and improved. An analysis based on percentages is supperficial. |
We have added a description of the relationship between practice and discourse to the section Theoretical discourses of repair. And we have added a description of our method to the Introduction section (lines 102-110). |

Reviewer 2 Report
This is an interesting study in which the authors analysed the main discourses about repairs in the posts of the largest Russian online communities to identify the environmental discourse. In this study, they identified four types of repair discourses, namely, environmental, pedagogical, social, and discourse about the right to repair. The authors found that the content of online communities contained almost no mention of environmental discourse.
The paper structure is generally appropriate and has some points of strength, although some minor details need to be revised before acceptance:
1. The introduction section was well structured, the definition of the problem was clear, and the research contribution was well presented.
2. In the second part, the theoretical discourses on repairs were described on the basis of four key discourses on repairs, including the environmental discourse, the pedagogical discourse, the social discourse, and the discourse on the right to repair. It is suggested that at the end of the theory of the four key discourses presented, the research questions are presented by the authors. After the presented theory and literature review, what are they looking for in this article?
3. The research method was clearly explained in the third part, the data collection, the creation of keyword lists and data preprocessing.
4. The other parts of the article were meticulously organised and I would just like to point out that it is better to move the section on limitations and future research after the conclusion.
Author Response
|
Reviewer 2. |
|
|
After the presented theory and literature review, what are they looking for in this article? |
Recognizing the importance of repair as a way to solve environmental problems, we analyze how the repair practice is mentioned in online repair communities to suggest ways to scale it as a green practice. We have added this sentence after literature review (lines 185-187 of the new version of the article). |
|
The other parts of the article were meticulously organised and I would just like to point out that it is better to move the section on limitations and future research after the conclusion. |
We have no objection to this proposal and have moved the section after the conclusion. |

Reviewer 3 Report
This article analyzes the dialogs among the Repair Online Community in Russia. The article provides very interesting findings. The applied methodology to reach the findings is very proper and the discussion of the findings is well provided. However, the main weakness of this article is that it fails to justify the importance and necessity of such (i.e., this) study. The authors should show it in two sections of the article: 1) introduction, and 2) discussion. In the introduction section the authors must provide evidence (e.g., statistics, or referring to literatures) to prove there is a research problem and it is very necessity to address this research problem, and at the end of the introduction, they must write a paragraph about the contributions of this study. On the other hand, in the discussion section, after discussing the findings, they must justify the contributions of the study based on the findings (for example by comparing the findings with the literature). The contributions should clearly say what are the novelties of this work, who and how can use the findings of this article.
The authors only referred to this matter in lines 449-450: “This finding can be used by leaders of online repair communities and eco-activists who will prepare posts for online repair communities.” That is not enough.
Author Response
|
Reviewer 3. |
|
|
However, the main weakness of this article is that it fails to justify the importance and necessity of such (i.e., this) study. The authors should show it in two sections of the article: 1) introduction, and 2) discussion. In the introduction section the authors must provide evidence (e.g., statistics, or referring to literatures) to prove there is a research problem and it is very necessity to address this research problem, and at the end of the introduction, they must write a paragraph about the contributions of this study. |
We confirmed the existence of a social problem based on the results of our previous studies (lines 81-85). In addition, we added the following part to the introduction: Thus, we have previously found that repair practices are rarely mentioned in online environmental communities in Russia, and repair activities involve only a few people. Recognizing the importance of repair as a way to solve environmental problems and finding a lack of references to repair as a green practice, we analyze how repair practice exists now and suggest ways to scale it (lines 85-89). |
|
On the other hand, in the discussion section, after discussing the findings, they must justify the contributions of the study based on the findings (for example by comparing the findings with the literature). |
The results of the study are explained in lines 384-391 of the new version of the article, they are compared with the results of other studies in lines 400-421. The significance of the results obtained is described in the Conclusion section. |
|
The contributions should clearly say what are the novelties of this work, who and how can use the findings of this article. |
We have expanded on this topic in lines 474-482. The research results can be used to implement strategies for scaling up the repair practice as a green practice. On the one hand, the results obtained can be the basis for the authorities’ decisions on greening social development. In addition, the results obtained can improve existing programs to support grassroots initiatives by focusing on events such as repair cafes. On the other hand, the actions of eco-activists can be adjusted according to our recommendations. For example, the results of the study will help to engage the leaders of online repair communities in promoting the environmental agenda. |

Reviewer 4 Report
Authors are presenting a very interesting argument about the missing environmental discourse from the promotion to repair. The gap they have identified in this manuscript is important.
My only suggestion is to improve the wording in the title. It would be great if you can indicate in the title that the research is about "appliances/equipment," as the word repair is usually understood in a broader way.
I detected a few instance in the paper where the authors have deviated from the manuscript style used a different writing style such as ones that you see in books. Use of footnotes is one example. How Kennedy & Givens citation has been used in the end of section 5 is another example. It would be neat to see them cleaned up to bring the writing more closer to the rest of the paper.
Author Response
|
Reviewer 4. |
|
|
My only suggestion is to improve the wording in the title. It would be great if you can indicate in the title that the research is about "appliances/equipment," as the word repair is usually understood in a broader way. |
The type of repair that we are writing about in the article is repair of household appliances, IT and telecommunications equipment. Following your suggestion, we have changed the title of the article. Online equipment repair community in Russia: searching for environmental discourse |

Round 2
Reviewer 1 Report
In my opinion, the article wasn't improved significantly. As I told in my first evaluation the hypothesis has to be rethought and come up with practical solutions.
Author Response
Dear reviewer, thank you for your thoughtful attitude to our article and for your suggestion of methodological approaches to the analysis of repair as a green social practice.
The first step towards studying repair as a green practice for us is to study how people involved in repair understand the motives of their participation, which topics within this social practice are attractive to them. To achieve our goal, we have chosen the method of discourse analysis as one of the ways to study social practices. It is widely used to study various social groups (for example, Pota M. et al. Multilingual evaluation of pre-processing for BERT-based sentiment analysis of tweets //Expert Systems with Applications. – 2021. – Т. 181. – С. 115-119; Muhammad, I., Mohd Hasnu, N.N., Ekins, P. (2021) Empirical Research of Public Acceptance on Environmental Tax: A Systematic Literature Review. //Environments, 8, 109) or the attitude of society to any phenomena (for example, Dayrell, C. (2019) Discourses around climate change in Brazilian newspapers: 2003-2013//Discourse & Communication. 13 (2) , pp.149-171; Koppenborg, F and Hanssen, U. 2021 Japan's Climate Change Discourse: Toward Climate Securitisation?// Politics and Governance. 9 (4) , pp.53-64; van Eck, C.W., Feindt, P.H. (2021) Parallel routes from Copenhagen to Paris: climate discourse in climate sceptic anti climate activist blogs // Journal of Environmental Policy & Planning. 24(5):1-16). Discourse analysis is also used when studying repair practices by other authors (for example, Graziano, V.; Trogal, K. The politics of collective repair: Examining object-relations in a postwork society. Cultural Studies 2017, 526 31, 634–658). In our opinion, this way is quite appropriate to achieve our research goals. In our case, this way involves the study of topics by keywords, which are indicators of the representation of a particular thematic focus – discourse. This method is also appropriate because we study posts in an online community where people are united by common topics for discussion.
In the future, it is possible to use other methods, for example, ethnography, in order to better understand those «who tinker, scavenge, save, buy used and give away to examine these practices in social context, lived experience and as embedded within larger political and economic structures of capitalist accumulation and abandonment» (Isenhour, C., & Reno, J. (2019). On Materiality and Meaning: Ethnographic Engagements with Reuse, Repair & Care. Worldwide Waste: Journal of Interdisciplinary Studies, 2(1), 1).
The theoretical framework for studying a phenomenon, for example, repair, can also be different. For example, Isenhour, C. and Reno, J. highlight elements of repair practice such as revaluation, resistance, care, relationality and reproduction and they study them through analysis interviews and observation (Isenhour, C., & Reno, J. (2019). On Materiality and Meaning: Ethnographic Engagements with Reuse, Repair & Care. Worldwide Waste: Journal of Interdisciplinary Studies, 2(1), 1). Therefore, when at the next step of our research we will study the practice of repair as a network of actors, your proposal to study global chains and interactions as elements of a circular economy (prevent-reuse-recycle-recover-dispose pattern) may be very heuristic for us. But now we would like to choose discourse analysis as a way.
We thank you for your valuable contribution to our research. If you think we misunderstood you, could you give us more information about what exactly we should change.